# Measurement Properties of Wearable Kinematic-Based Data Collection Systems to Evaluate Ball Kicking in Soccer: A Systematic Review with Evidence Gap Map [note 1]

**DOI:** 10.3390/s24247912

**Published:** 2024-12-11

**Authors:** Luiz H. Palucci Vieira, Filipe M. Clemente, Rui M. Silva, Kelly R. Vargas-Villafuerte, Felipe P. Carpes

**Affiliations:** 1Grupo de Investigación en Tecnología Aplicada a la Seguridad Ocupacional, Desempeño y Calidad de Vida (GiTaSyC), Facultad de Ingeniería y Arquitectura, Campus Callao, Universidad César Vallejo (UCV), Callao 07001, Lima, Peru; 2Escola Superior Desporto e Lazer, Instituto Politécnico de Viana do Castelo, Rua Escola Industrial e Comercial de Nun’Álvares, 4900-347 Viana do Castelo, Portugal; filipe.clemente5@gmail.com (F.M.C.); rui.s@ipvc.pt (R.M.S.); 3Sport Physical Activity and Health Research & Innovation Center, 4900-347 Viana do Castelo, Portugal; 4Department of Biomechanics and Sport Engineering, Gdansk University of Physical Education and Sport, 80-336 Gdańsk, Poland; 5Facultad de Ciencias de la Salud, Campus Callao, Universidad César Vallejo (UCV), Callao 07001, Lima, Peru; kvargasvi@ucv.edu.pe; 6Applied Neuromechanics Research Group, Laboratory of Neuromechanics, Federal University of Pampa (Unipampa), P.O. Box 118, Uruguaiana 97500-970, RS, Brazil; carpes@unipampa.edu.br

**Keywords:** sports engineering, quality control, measurement error, technique, football, human movement, validity, reliability, accuracy, performance analysis

## Abstract

Kinematic assessment of ball kicking may require significant human effort (e.g., traditional vision-based tracking systems). Wearables offer a potential solution to reduce processing time. This systematic review collated measurement properties (validity, reliability, and/or accuracy) of wearable kinematic-based technology systems used to evaluate soccer kicking. Seven databases were searched for studies published on or before April 2024. The protocol was previously published and followed the PRISMA 2020 statement. The data items included any validity, reliability, and/or accuracy measurements extracted from the selected articles. Twelve articles (1011 participants) were included in the qualitative synthesis, showing generally (92%) moderate methodological quality. The authors claimed validity (e.g., concurrent) in seven of the eight studies found on the topic, reliability in two of three, and accuracy (event detection) in three of three studies. The synthesis method indicated moderate evidence for the concurrent validity of the MPU-9150/ICM-20649 InvenSense and PlayerMaker™ devices. However, limited to no evidence was identified across studies when considering wearable devices/systems, measurement properties, and specific outcome variables. To conclude, there is a knowledge base that may support the implementation of wearables to assess ball kicking in soccer practice, while future research should further evaluate the measurement properties to attempt to reach a strong evidence level.

## 1. Introduction

In the sports engineering field, particularly in the context of team ball sports, one of the main goals is to conceive monitoring tools that can be used in naturally occurring environments [1,2]. This is given the general poor construct validity of laboratory or even field-based tests outside game-play to evaluate skill-related performance, such as in the case of soccer kicking (i.e., passing or shooting) ability [3,4,5,6]. Testing and reporting the soccer kicking skill features (e.g., lower limb movement pattern and velocity) to coaching staff is of paramount importance since (a) it could help provide feedback aiming at perfecting players’ movements [1], (b) there are strong relationships between kicker mechanical parameters and kick outcomes [7,8,9,10], and (c) the number of injury occurrences (e.g., >10–15%) associated with ball-kicking events is not negligible [11,12,13], which in turn may be prevented with adequate monitoring of player motions under training or match conditions. In this sense, wearable technologies have been recognized to assist in biofeedback purposes [14]. Indeed, both the use/perception of the importance of wearables and the monitoring of key performance indicators derived from ball-kicking features (e.g., shooting or passing metrics) are frequent in soccer contexts [15,16].

To collect data on position, velocity, and acceleration during kicking movement, there are various electronic performance and tracking systems (EPTSs) currently commercially available, such as optical-based, inertial measurement units (IMUs), GNSS, and LPS [17,18,19]. High-speed semi-automatic three-dimensional optical systems are generally considered the “gold standard” given their lowest levels of reported measurement error [18,20]. However, these systems are limited in capture volumes and traditionally require reflective markers attached to players’ bodies [17] and can also be labor-intensive. Consequently, the use of this method during official competition is therefore limited. In this sense, artificial intelligence-assisted markerless video-based tracking could be a possible solution, representing a less invasive method for capturing kicking data [1,21]. Aside from the advantages of such a straightforward approach, the calibration of extended volumes is challenging, potential occlusion effects can still occur, and it is necessary to position the cameras in optimal locations (e.g., high). Once again, this can collectively constrain the use of optical-based systems when investigating in-game ball kicking in soccer.

Since the mid-2010s, the Fédération Internationale de Football Association (FIFA) has allowed the use of EPTSs [22]. Although biofeedback appears to be available in many of the current EPTSs, its definitive implementation in day-to-day practice has not yet been widely observed, as in the case of soccer kick testing/monitoring [23]. IMUs (e.g., Xsens Technologies B.V., Enschede, The Netherlands) represent a wearable type of EPTS that may allow kicking kinematics to be obtained [24]. In fact, wearables may assist in overcoming many of the limitations mentioned above associated with the use of video-based techniques, also helping in possibly providing real-time data [25]. On the other hand, not all of them are feasible to apply in competition settings. Examples included those multi-sensor apparatuses requiring unique clothing and/or reinforced attachment on various body locations, which may potentially constrain movement patterns. Starting from September 2023, a specific wearable foot-mounted apparatus that enables tracking lower extremity movements was the first of the category approved by FIFA, the so-called “PLAYERMAKER” system (Playermaker LTD) [26,27]. Of note is that there are recent criticisms regarding some aspects of FIFA’s quality performance reports for EPTSs [28,29].

While several previous literature reviews addressed methodological details of wearables applied to kinematic analysis in sports tasks [25,30,31,32,33,34,35,36,37,38,39], none have solely examined soccer kicking skills. Thus, a systematic analysis of the level of scientific evidence available concerning the specific measurement properties of wearables used in monitoring ball-kicking actions is still lacking. One particular concern when assessing soccer kicking movement kinematics refers to the potential distortion in time-series data caused by the impact of the foot with the ball [40,41]. Also, ball kicking may involve a very rapid change in speed values of the lower extremity from preparation to follow through. The concomitant demands upon velocity and accuracy constraints, which are known to elicit paradoxical central inputs of players [42], probably tend to contribute to producing a substantial degree of variability in the assessment of soccer kicking as well [43]. This emphasizes the need to choose the methods for collecting/processing kick data carefully. Taken together, there is an evident necessity to collate studies that have determined measurement errors of wearable technology devices used to measure biomechanical aspects of soccer kicking as this can help inform practitioners, researchers, and industry workers in the area. Therefore, the present study aims to synthesize evidence from scientific journal articles that provided data about measurement properties (validity, reliability, and/or accuracy) of wearable kinematic-based technology systems used to evaluate ball kicking in the soccer context through a systematic literature review.

## 2. Methods

The protocol established for the present systematic review was previously registered (OSF REGISTRIES—project ZM3J6) and is available in full elsewhere [44]. The Institutional Research Ethics Committee, from the Universidad César Vallejo Perú, granted permission to run this review study. The review protocol was established considering the items from the PRISMA 2020 statement (Appendix A) [45].

### 2.1. Database Searches

Table 1 contains terms used for the searches across the databases selected. For the formulation of the search string, terms between the same column were combined with the Boolean operator OR, while the groups of terms between columns were combined with the Boolean operator AND. The searches were performed on 23 April 2024, through seven electronic databases: EBSCOHost, IEEE Xplore, Physical Therapy and Sports Medicine, ProQuest, PubMed, Scopus, and Web of Science.

The selected search terms attempted to respect the PICO headings method [46]. To assist in managing titles across the whole review process (i.e., from initial searches until the step of final inclusion), Zotero software was used in the present study (v6.0.30; https://www.zotero.org/; accessed on 5 February 2024). The researchers involved in each aspect of the development of all the review steps are described in the protocol study [44].

### 2.2. Eligibility Criteria

#### 2.2.1. Inclusion Criteria

Studies were included (i) when published (even ahead of print) as original research articles (ii) in scientific peer-reviewed journals, (iii) having their abstracts available for screening in the respective databases, (iv) with full text available—in the English language—for download (i.e., full text download button available for the record in the respective database), and (v) describing aspects related to research ethics with human participants [47] or respective exemption to perform the investigation [48]. The inclusion criteria also included (vi) participants as soccer players and/or human data collected in a football-related environment, (vii) observation and/or experimental protocol including ball kicking (passing or shooting) while (viii) participants used at least one wearable kinematic-based device/system that was (ix) evaluated concerning its measurement properties (i.e., validity and/or reliability and/or accuracy aspects).

#### 2.2.2. Exclusion Criteria

Studies were excluded if they (i) were published in the form of grey literature, (ii) were retracted, (iii) investigated only the application/feasibility aspects of wearables, (iv) assessed only movements pertaining to football codes other than ball kicking, (v) were explicitly conducted in football codes other than soccer, (vi) reported observations and/or experimental protocols where a ball was not kicked, (vii) reported results only concerning ball motion/flight, and (viii) were full texts with no information on the location to which the wearable was attached to the body of participants [44].

### 2.3. Data Extraction

Using custom-prepared Microsoft^®^ Excel spreadsheets (v.Professional Plus 2016, Microsoft Corporation, Redmond, USA), the following indicators were extracted from the included studies, according to their specific aims [49,50]: (i) validity measures [e.g., correlation coefficient, root mean square difference (RMSD), and area under the receiver operating curve (AUC)]; (ii) reliability measures [e.g., intraclass correlation coefficient (ICC) and coefficient of variation (CV)], and/or accuracy measures (e.g., F1 score, prediction accuracy, and percentage of correctly classified activity). For all studies, the main characteristics of methods were also extracted, including the wearable (number of sensors, brand of the system/device used, location attached to the body, and acquisition frequency), metrics collected by the wearable, number of participants and sample profile (sex, age, playing level, and positional role), experimental protocol (aspects of testing/observation of ball kicking, duration, condition(s), and lower limb considered), form of data analysis, statistical outputs, and summary of findings/conclusions.

### 2.4. Methodological Quality and Risk of Bias Assessments

Here, methodological quality was assessed using the COSMIN checklist or the QUADAS-2 tool, depending on the measurement property addressed by each study. In particular, forms H and B from the COSMIN checklist were adopted to make judgments, respectively, on the quality of studies that evaluated validity and reliability aspects [51,52]. These studies were rated in each domain of the mentioned tool using a 4-point scale (excellent, good, fair, or poor) [53]. Subsequently, the ratings for each question were converted into values (excellent = 3 points, good = 2 points, fair = 1 point, and poor = 0 points), and then punctuations of all questions were summed for each study. For validity aspects, studies were deemed as having high quality (∑ = 14–18), moderate quality (∑ = 8–13), or low quality (∑ = 3–7). Likewise, for reliability aspects, studies were deemed as having high quality (∑ = 25–33), moderate quality (∑ = 14–24), or low quality (∑ = 3–13). For studies assessing accuracy aspects, the QUADAS-2 tool was adopted [54]. These studies were rated in each domain of the mentioned tool using a 3-point scale (low = 2 points, unclear = 1 point, high = 0 points) and then summed for each study. For accuracy aspects, studies were deemed as having high quality (∑ = 11–14), moderate quality (∑ = 8–10), or low quality (∑ = 0–7).

The risk of bias in results or inferences was determined using the RoBANS tool [55]. Each study was analyzed individually and rated as presenting a low, high, or unclear risk of bias concerning the selection of participants, confounding variables, measurement of exposure, blinding of outcome assessments, incomplete outcome data, and selective outcome reporting. RevMan software (v5.3; The Cochrane Collaboration, Denmark) was used to compute graphs of the resulting risk of bias. If any discrepancy existed in the rating between the two evaluators involved—in evaluating methodological quality and risk of bias in the included studies—a third (senior) researcher was then consulted until a consensus was reached.

### 2.5. Evidence Synthesis

To provide a synthesis of findings across the studies, the level of evidence was classified as (a) strong [consistent findings (≥75% of the studies reported results in the same direction) across multiple high-quality studies], (b) moderate (consistent findings across multiple moderate-quality and/or one high-quality study), (c) limited (consistent findings across one moderate-quality and/or only low-quality studies), (d) conflicting (<75% of studies reported results in the same direction), or (e) no evidence (no studies found). An evidence gap map was also constructed [44].

## 3. Results

A total of 676 title entries were initially identified when pooling results of the searches across all databases in addition to the titles manually identified. Following the elimination of duplications and low-relevant studies, the fields of title, abstract, and keywords of 80 studies were inspected. This procedure resulted in the exclusion of 44 studies. Finally, after reading the full texts of 36 studies assessed for eligibility and applying all the selection criteria, a total of 12 studies [56,57,58,59,60,61,62,63,64,65,66,67] addressing the measurement properties of wearable-based systems to evaluate soccer kicking features were included in the qualitative synthesis. Given the heterogeneity in methods observed across the included studies, a meta-analysis was not possible in the present review. Eight studies evaluated the respective validity aspects [56,57,61,62,63,64,66,67], three evaluated reliability [58,62,63], and three evaluated accuracy [59,60,65]. Figure 1 depicts the frequency of studies included according to the year of publication. Reasons for exclusions in the final eligibility stage included, for example, studies available only as conference proceedings, reporting results of kicking aggregated with other actions, or lacking a statement regarding an ethics committee (see Figure 2).

### 3.1. Characteristics of the Included Studies 

The main aspects of the methods in the included studies are summarized in Table 2. The total number of participants was *n* = 1011 for all studies pooled (median = 11 participants). Seven studies included only male participants (58%) [56,57,59,62,63,64,66], four included both men and women (33%) [58,60,61,65], and in one, this information was unclear [67]. Eight studies were conducted with adult samples (67%) [56,57,58,59,62,63,64,66], two with youth (17%) [60,61], and one with both youth and adults [65]. One study included elite players [62], one both elite and sub-elite [56], and the remainder sub-elite, except two studies in which unclear information was provided [64,67].

Two studies reported the inclusion of samples of participants from all playing positions (pooled) [56,62], while the information related to the positional role of participants was unclear in all the remaining studies (83%). Common metrics collected by the wearables included velocity [pelvis [57] and foot (25%; [57,62,67])], acceleration [pelvis [59,66], thigh [59,66], shank [58,59,66], ankle [60,61], and foot (33%; [62,63,65,67])], angular displacement (pelvis, shank [57], and hip and knee [64]), angular velocity (pelvis [59,66], hip [64], knee [56,57,64], thigh [57,59,66], shank [57,59,66], ankle [62,63], and foot [65,67]), and angular acceleration of the hip and knee [56].

The number of sensors used in the data collections ranged from one sensor [67] to two (33%; [58,62,63,65]), four [60], five (42%; [56,59,61,64,66]), and seventeen sensors [57]. Placement locations included the waist [61], pelvis (33%; [57,59,64,66]), hip [60], thigh (42%; [56,57,59,64,66]), shank (50%; [56,57,58,59,64,66]), ankle (17%; [60,61]), and foot (42%; [57,62,63,65,67]). Approximately one-quarter of the studies analyzed preferred limbs [56,57,58], while half assessed both preferred and non-preferred limbs [59,62,63,64,65,66]. The information related to functional lateralization was unclear in three studies [60,61,67]. In addition, some studies also placed devices on the head, upper arm, forearm, sternum, shoulder, hand [57], wrist [61], and lower back [56].

Wearable devices/systems used in the included studies consisted of IMUs MPU-9150 (InvenSense) [56,59,62,63,66], ICM-20649 (InvenSense) [64,67], MVN Link (Xsens Technologies) [57], VICON Blue Trident (IMeasureU) [58], PlayerMaker™ [62,63], and accelerometers GT1M/GT3X (ActiGraph) [60] and GENEActiv (Activinsights) [61]. The information regarding the wearable devices/systems used was unclear in one study [65]. Acquisition frequencies ranged from 80 Hz [61], 100 Hz [67], 200 Hz [65], 240 Hz [57], 250 Hz [64], 500 Hz [56,59,66], 1000 Hz [63], to 1600 Hz [58]. Two studies presented unclear information relating to acquisition frequency [60,62].

The time of testing across studies varied from 5 min [58] to 10 min [56] and 15 min of warming-up, plus the experimental protocol [66]. One study provided 15 min of familiarization plus the experimental protocol [62]. Three studies provided descriptions of the duration of the protocol, consisting of 5 min [61], 20 min [60], and 35 ± 17 to 73 ± 38 min [65]. Unclear information about the total time duration of the warm-up and/or experimental protocol was observed in five studies (~42%; [57,59,63,64,67]), for which information on the composition of the experimental protocol, but not timing, was reported. The description of protocols adopted in all studies regarding the ball-kicking action is also presented in Table 2.

Ball kicking was collected as passing in eight studies (67%; [58,59,60,61,63,64,65,66]) and shooting in seven studies (58%; [56,57,59,64,65,66,67]). Unclear information about whether ball kicking was collected as a pass or shot was presented in one study [62]. Finally, most studies (92%) used protocols consisting of performing ball kicking as closed skill/constrained tasks [56,57,58,59,60,61,62,63,64,66,67]. There was only one exception in a study that considered data in both conditions, i.e., closed skill/constrained tasks and also open skill/unconstrained tasks format [65].

### 3.2. Measurement Properties

The main results regarding measurement properties (validity, reliability, and accuracy) presented by the wearable kinematic-based data collection systems to evaluate ball kicking in soccer are summarized in Table 3. The table also contains the summary—main conclusions—provided by the authors of each individual included study. In general, the validity of wearables was claimed by the authors in seven out of eight studies [56,57,62,63,64,66,67], reliability in two out of three studies [62,63], and accuracy in three out of three studies [59,60,65].

### 3.3. Methodological Quality and Risk of Bias in the Included Studies

Table 4 shows the results of the methodological quality assessment for the eight studies that addressed validity aspects. All studies showed moderate quality. Item 3 (sample size) presented a higher number of studies scoring “poor” (100%) followed by item 6 (38% of studies on this topic lacked correlations or AUC calculations).

Table 5 shows the results of the methodological quality assessment for the three studies that addressed reliability aspects. Once again, all studies showed moderate quality. Item 3 (sample size) presented a higher number of studies scoring “poor”, followed by item 11 (67% of studies on this topic lacked ICC calculations).

Table 6 shows the results of the methodological quality assessment for the three studies that addressed accuracy aspects. Two studies had moderate methodological quality while one study showed low methodological quality (33%). Item A from domain 1 (patient selection) showed the high number of studies rated with the poorest score (100% with a high risk of bias relating to the selection of participants).

Figure 3 and Figure 4 contain, respectively, the results of the risk of bias for individual studies and a summary of the results for all studies pooled. The item selection of participants presented a higher number of studies rated as high risk (17%). The item blinding of the outcome assessment showed a higher number of studies rated as unclear risk (75%).

### 3.4. Summary of Evidence

Figure 5 depicts the interaction between measurement properties and kinematic data collected by wearables in the studies included. Open circles are the number of studies. Thresholds for qualitative synthesis were applied according to those described previously (Section 2.5). Regardless of the outcome variable, taking into account the wearable devices/systems and the measurement properties (Figure 6), the MPU-9150 (InvenSense) showed moderate evidence for concurrent validity across studies [62,63,66], limited evidence for discriminative validity [56], within-session reliability [62], intra-unit reliability [63], and event detection accuracy [59], and no evidence for between-session reliability. The ICM-20649 (InvenSense) showed moderate evidence for concurrent validity across studies [64,67] and no evidence for the remainder of measurement properties. The PlayerMaker™ showed moderate evidence for concurrent validity across studies [62,63], limited evidence for within-session reliability [62] and intra-unit reliability [63], and no evidence for the remainder of the measurement properties. All the other devices/systems (MVN Link-Xsens Technologies [57], VICON Blue Trident-IMeasureU [58], GT1M/GT3X-ActiGraph [60], and GENEActiv-Activinsights [61]) showed limited to no evidence for the measurement properties. Finally, limited to no evidence was also identified across studies when considering wearable devices/systems, measurement properties, and specific outcome variables.

## 4. Discussion

The present systematic review aimed to synthesize the evidence on the measurement properties (i.e., validity, reliability, and/or accuracy) of wearable kinematic-based systems used for evaluating ball kicking in soccer. While a meta-analysis was not viable in the present investigation due to variations in various aspects of methods across the included studies and the generally reduced number of available published studies on the subject, the best evidence synthesis and gap map methods allowed us to draw some inferences. The main findings revealed moderate evidence supporting the concurrent validity of wearable systems, including the (i) MPU-9150, (ii) ICM-20649—both i and ii pertaining to InvenSense (San Jose, CA, USA)—and (iii) PlayerMaker™ (Tel Aviv, Israel), for soccer kicking analysis in controlled settings. However, evidence for between-session reliability and accuracy remains limited, particularly in real-game scenarios. The current study is a revised and expanded version of a previous conference paper [68].

The studies included predominantly focused on the validity of wearable systems, with a significant proportion showing its concurrent validity for different measures (e.g., joint angles, velocity, and acceleration). The moderate evidence supporting the concurrent validity of the wearable devices MPU-9150, ICM-20649, and PlayerMaker™ in controlled environments [62,63,64,67] suggests a great capability for these wearables to measure kinematic variables with a degree of accuracy that aligns with the gold-standard instruments for the same purpose. Regarding other wearables, such as the MVN Link system, Blair et al. [57] suggested that the MVN Link system has concurrent validity compared to a high-standard motion analysis system. However, the high error levels observed in faster kicking attempts indicate that while the system can detect movements, the precision of this system might be velocity-dependent. Moreover, limited evidence was found for devices like the GT1M/GT3X and GENEActiv. Their use in assessing kicking kinematics was less straightforward, showing moderate correlations for certain activities but not specifically highlighting their effectiveness for detailed kinematic kicking analysis [61]. The low cross-validity (i.e., for the GENEActiv) indicates that devices might be better suited for general activity recognition rather than precise kinematic measurements in particular movements or tasks. Most of these devices were not specifically designed for the high-speed, high-impact nature of soccer kicking, which may explain the limited evidence in kicking kinematic analysis.

The capability of these wearables to measure kinematic variables with a degree of accuracy that aligns with the gold-standard instruments is crucial as it benchmarks the actual technology used against established methods, providing a basis for further exploration into their practical utility in sporting settings [69]. However, the inherent variability in soccer kicking velocity and accuracy [43,70,71,72,73,74] requires wearables that measure and interpret these actions considering their context. This fact is not uniquely challenging to soccer; similar demands for contextual data interpretation are also needed in other sports where precision and variability in movement are paramount (e.g., tennis serves [75]). Furthermore, the distortion of time-series data due to the impact of the foot with the ball also complicates this scenario of contextual factors [40,41]. Indeed, the variability in validity outcomes of the included studies in the present systematic review exposes the complexity of collecting biomechanical data in real-world sports settings [14,30,76].

The still limited evidence on reliability (e.g., between-session) and the event detection accuracy showed an important gap in the practical application of wearable technologies in the context of soccer kicking assessment. This is critical as reliability over time and across different conditions is essential for wearables to be integrated effectively into the training process to improve decision-making. Another systematic review that addressed the use of IMUs in volleyball also identified reliability results that were worse than validity ones [77]. The limited between-session reliability and accuracy in real-game scenarios found in the present systematic review might be attributed to (i) the possible negative results from the (unpublished) research carried out [78]; (ii) on the occasion that (i) is true, then there might be a need to improve the existing data/equipment capture/processing methods and/or (iii) differences in sensor placement, the number of sensors used, or the specific measures being measured (e.g., velocity, acceleration). Burland et al. [58] explored the reliability of the VICON Blue Trident (IMeasureU) system, finding moderate to high ICC values during kicking actions. However, the lower reliability for ball kicking compared to other actions assessed (i.e., locomotion tasks without ball handling) suggests that the nature of kicking might introduce variability that affects the device’s consistency. This could be due to the placement of sensors or the sensitivity to rapid changes in acceleration. Regardless of the measurement property, replication studies are encouraged using an adequate/sufficient number of participants [79] given that the included studies of the present systematic review showed, in general, low sample sizes. The same problem of a small number of participants across the literature studies was identified in a systematic review that covered the topic of wearables in basketball [38].

Studies utilizing foot-mounted sensors like the PlayerMaker^TM^ system have shown potential, yet the generalizability of these findings to other wearable configurations or all player positions remains uncertain. Specifically, the placement of sensors may directly affect data accuracy [80]. Foot-mounted sensors might excel in capturing kicking dynamics in controlled settings. Still, they might not generalize to real-world settings [81], such as the training context, where impacts and other inter-player contacts are very likely to occur. Thus, here, the presence of potential biases arising from experimental (e.g., controlled) conditions should be not overlooked. Using observations from well-defined ball kicking as closed skill/constrained tasks would possibly favor one outcome over others. Also, the need for regular calibration or updates to software algorithms to account for new data or different playing conditions is often overlooked, and reliability can be negatively affected [39]. Furthermore, as players fatigue, their biomechanics and outcomes change [82,83], potentially leading to different between-session data outputs from wearables [84]. This aspect of the soccer training context may explain the still limited evidence on the reliability and accuracy of data over multiple sessions. Moreover, accuracy in event detection, particularly for kicking, was not explicitly detailed for many of these devices. Notwithstanding, soccer kicking performance (e.g., ball release velocity) has been position-dependent in either youth/adult or men/women players [85,86,87]. In addition, in-game situations like ground duels could potentially create challenges for the signal processing of wearables in the soccer context and also vary according to playing position [88,89]. Thus, future analysis of measurement properties in wearables should consider the present results also separated by playing position. It is also advisable that the metrics included be justified in relation to their actual practical meaningfulness (e.g., importance to the monitoring of the performance of the kicking skill/overall player performance, age-related development, and/or injury risk mitigation).

The present systematic review has inherent limitations. Firstly, our focus was narrowed to the validity, reliability, and accuracy of wearable devices in soccer kicking actions, which might not capture the full capabilities of wearable technology applications in soccer or other sports. Although this seems to also be the case in other team sports (e.g., handball [90]), it is also necessary to recognize that in the analysis (i.e., qualitative synthesis) considering wearable devices/systems, measurement properties, and specific outcome variables, limited to no evidence was verified in the current literature. Another significant limitation is the demographic bias, where there was a predominance of men and adult participants in included studies, thereby limiting the generalizability of our findings across the broader populations. Using a traditional approach to systematically review the literature (e.g., excluding conference proceedings [91,92,93,94,95,96,97,98]) may have led to the overlooking of potentially worthwhile scientific findings in the synthesis presented. Our review’s lack of longitudinal studies highlights a gap in understanding how these technologies perform over time. It would be beneficial for future studies to adopt a broader inclusion criterion, encompassing studies that include other football codes to ensure that wearable technology’s benefits are applicable. For example, another potential limitation of the methods in the current review is that—in an attempt to collate specific evidence for the team sport of soccer—the realization of studies explicitly conducted in football codes other than soccer was established as an exclusion criterion. Also, ball-kicking outcomes are age-dependent [99,100,101,102]. In studies regarding the measurement property accuracy, only one used a sample of exclusively senior participants, while the other two used solely youth or mixed samples (adult and senior pooled). Extrapolating the measurement properties of studies with youth participants to seniors requires caution, and the opposite, too. The same thought can be derived regarding competitive standards; results encountered in non-elite levels may not be interchangeable with the elite due to the distinct kicking outputs according to playing standards [103,104,105]. In the present review, only two studies investigated the measurement properties of wearables in elite players, and their use in practice has become increasingly widespread in recent years among professional clubs. Moreover, the standardization of testing and calibration protocols for these devices could improve the consistency and comparability of the outcomes (e.g., using recommendations from the International Society of Biomechanics [106]). Finally, there is a need to test and report aspects related to the robustness of sensor solutions, their adaptation to player body segments, and resilience levels to cope with impacts and other inter-player contacts. Longitudinal research also considering the duration of official matches and training sessions should be prioritized to assess the long-term reliability and utility of wearable data.

## 5. Conclusions

From the present systematic review with the best evidence synthesis and evidence gap map methods, it is possible to conclude that there is a knowledge base that may support the implementation of wearables to assess ball kicking in practice. The wearable devices/systems that so far have the highest level of evidence—moderate—for the validity (i.e., concurrent) across the existing literature studies are the MPU-9150 (InvenSense), ICM-20649 (InvenSense), and PlayerMaker™ (Tel Aviv, Israel). In contrast, limited to no evidence is noted regardless of the wearable devices/systems considered regarding both reliability (e.g., between-session) and accuracy (event detection) aspects. Also, the conclusions are applicable to a greater extent to male adult players. Future research should further evaluate the measurement properties of wearables to attempt to reach a strong evidence level regarding their performance in assessing ball kicking in soccer, mainly considering the inclusion of a sufficient number of players, under-represented populations (e.g., female and youth players), and data collection approaches during actual training sessions and official matches.

## Figures and Tables

**Figure 1 sensors-24-07912-f001:**
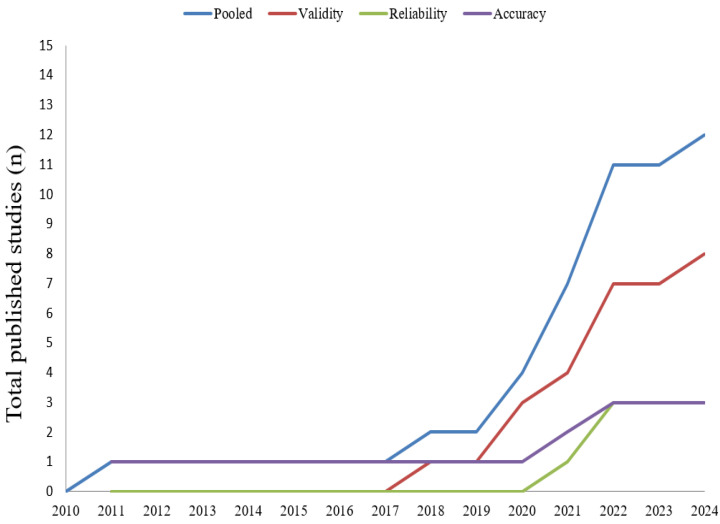
Description of the number of included studies according to the publication year.

**Figure 2 sensors-24-07912-f002:**
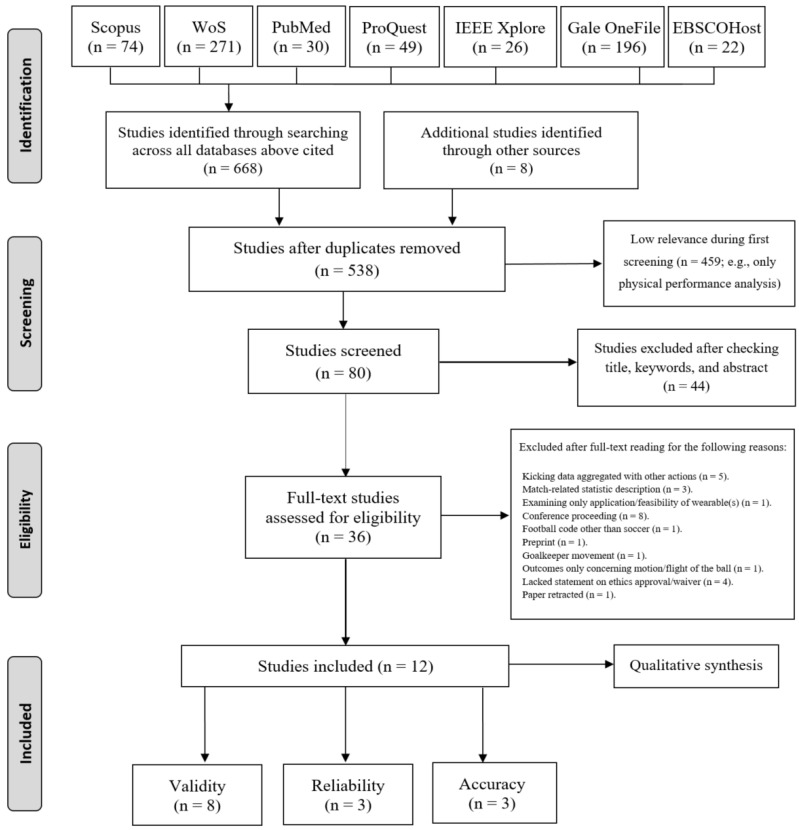
PRISMA flow diagram.

**Figure 3 sensors-24-07912-f003:**
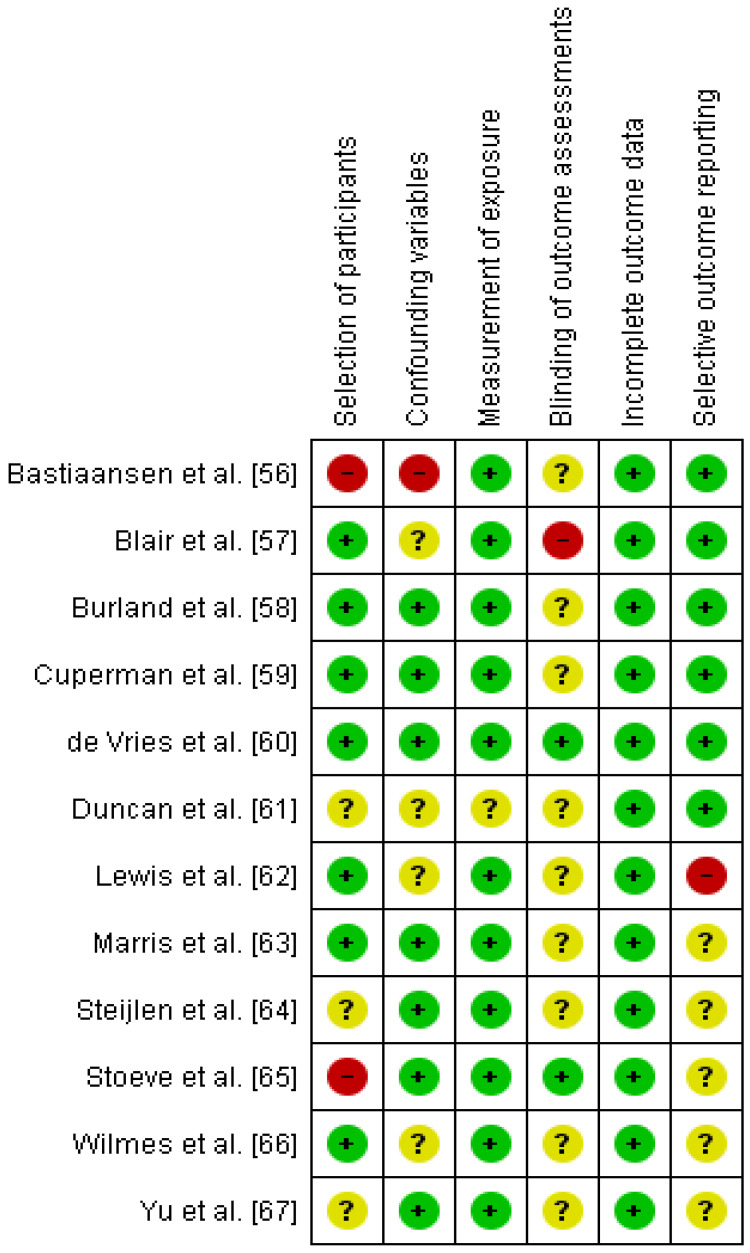
Judgments made about each risk of bias item for each included study [56,57,58,59,60,61,62,63,64,65,66,67]. (+) Low risk. (–) High risk. (?) Unclear risk.

**Figure 4 sensors-24-07912-f004:**
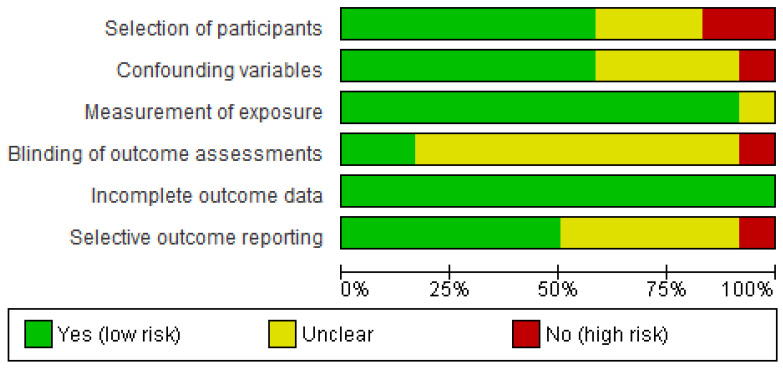
Risk of bias for each item as percentages across all the included studies.

**Figure 5 sensors-24-07912-f005:**
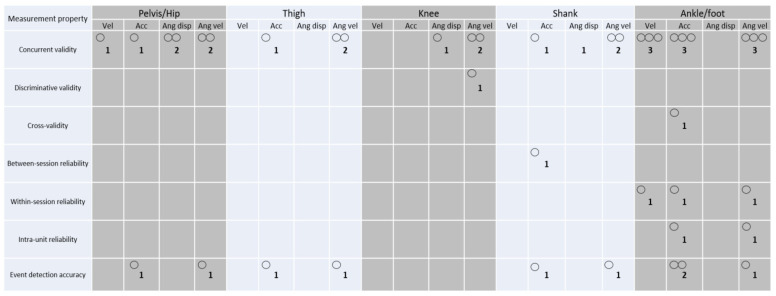
Interaction between the measurement properties tested and kinematic data collected by the wearables. In the figures, open circles represent the number of studies included and vel: velocity; acc: acceleration; ang: angular; and disp: displacement.

**Figure 6 sensors-24-07912-f006:**
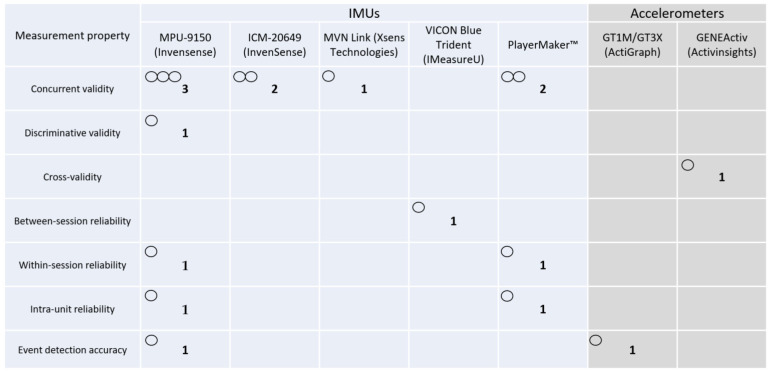
Evidence gap map regarding the measurement properties tested and the wearable devices/systems used. Open circles represent the number of studies found.

**Table 1 sensors-24-07912-t001:** Search terms used for the current systematic review through a PICO framework.

Population	Intervention	Comparison	Outcome
soccer	wearable *	validity	kick *
football *	inertial measurement unit	reliability	shoot *
association football	IMU	measurement error	pass *
11-a-side	acceleromet *	accuracy	skill
	microtechnology	precision	technical
	micro-electrical mechanical system		
	MEMS		
	global positioning system		
	global navigation satellite system		
	local positioning system		
	GPS		
	GNSS		
	LPS		

* Wildcard term.

**Table 2 sensors-24-07912-t002:** Some characteristics extracted from the methods used in the studies included in the current review.

Reference	Metrics Collected	Sex	Age	Level	*N*	Positional Roles	Tested			Wearable Device/System	Frequency	Experimental Protocol	Form of Data Analysis	Validity Results	Reliability Results	Accuracy Results
							Validity	Reliability	Accuracy							
Bastiaansen et al. [56]	Hip load; knee load (derived from joint angular accelerations); and knee extension velocity	Men	Adult	Elite; sub-elite	28	All (pooled)	✓			IMUs (MPU-9150, InvenSense, San Jose, CA, USA)	500 Hz	Five 5 m maximal instep kicks aiming at a full-sized goal without accuracy demands	Data compared across playing levels	MANOVA	--	--
Blair et al. [57]	Foot speed; pelvis velocity; shank, thigh, and knee (angular velocity); shank and pelvis (sagittal angle) at impact; and max support-knee extension, min kick-leg knee angle, and max hip extension	Men	Adult	Sub-elite	10	--	✓			IMUs (MVN Link, Xsens Technologies B.V., Enschede, The Netherlands)	240 Hz	Five 12 m instep kicks, five 12 m inside kicks, five 20 m instep kicks, and five maximal instep kicks	Data compared against a 12-camera motion analysis system (T-40 series, Vicon Nexus v2, Oxford, UK)	GLMM; variances; error; Studentized residual vs. predicted plots	--	--
Burland et al. [58]	Tibial acceleration value; cumulative impact load; total steps; and cumulative bone stimulus (number of steps multiplied by the tibial acceleration)	Men; women	Adult	Sub-elite	10	--		✓		VICON IMeasureU Blue Trident dual-g sensors (IMeasureU, Auckland, New Zealand)	1600 Hz	Three trials passing the ball forward at a self-selected pace	Data analyzed across three test sessions at least 7–10 days apart	--	ICC	--
Cuperman et al. [59]	Triaxial accelerations and angular velocities of pelvis, thigh, and shank	Men	Adult	Sub-elite	11	--			✓	IMUs (Ivensense MPU-9150)	500 Hz	Football-related activities, including passes, shots, jumps, and sprints, among others, to simulate an actual football match	Data compared against a manually annotated category (activity)	--	--	F1 score Prediction Accuracy
de Vries et al. [60]	Accelerometer data (counts) of the ankle	Men; women	Youth	Sub-elite	58	--			✓	ActiGraph accelerometers (GT1M/GT3X; ActiGraph, Pensacola, FL, USA)	--	Monitored 20 min activities including sitting, standing, walking, running, rope skipping, kicking the ball, and cycling	Data compared against research assistant recordings	--	--	%Correctly classified activity Contingency tables
Duncan et al. [61]	Triaxial accelerometry data for the ankle	Men; women	Youth	Sub-elite	20	--	✓			GENEActiv accelerometers (Activinsights, Cambridge, UK)	80 Hz	Monitored 5 min for each activity with 5 min resting: lying supine, standing, running, and instep passing a football (size 3 ball, over a distance of 5 m at a cadence of 10 and 20 passes/min) and dribbling	Data correlated with metabolic equivalents; compared among activity types	rho-Spearman; ROC curve AUC Sensitivity Specificity	--	--
Lewis et al. [62]	Velocity of the foot—or ball release velocity	Men	Adult	Elite	4	All (pooled)	✓	✓		IMUs (PlayerMaker™, Tel Aviv, Israel; including two components from MPU-9150, InvenSense, CA, USA)	--	Twelve kicks in a static ball (six with each foot) at low, moderate, or high subjective intensities with foot region self-selected	Data correlated with joint angular velocity from a high-speed camera system (Quintic Consultancy Ltd., Sutton Coldfield, UK); data analyzed within session for each subjective intensity	r-Pearson	CV	--
Marris et al. [63]	Ball touch and release occurrences derived from accelerometer and gyroscope traces	Men	Adult	Sub-elite	12	--	✓	✓		IMUs (PlayerMaker™, Tel Aviv, Israel; including two components from MPU-9150, InvenSense, CA, USA)	1000 Hz	A total of 8640 ball touches and 5760 releases in technical soccer tasks where a given player served the ball to another, considering two pre-determined distances (13.2 and 18.7 m)	Data contrasted with manual coding/video (SportsCode Elite, v. 11.2.23, SportsTec, Warriewood, Australia); data analyzed across three codings made by the performance analyst concerning a same specific subset of tasks	Proportion of agreement	CV	--
Steijlen et al. [64]	Hip and knee joint angles and angular velocities	Men	Adult	--	1	--	✓			IMUs ICM-20649 (InvenSense, San Jose, CA, USA)	250 Hz	Three trials kicking a ball preceded by a few steps, at three different intensities (50, 80, and 100% of maximum effort)	Data compared against and correlated with eight cameras (Vicon V5 cameras, Vicon Motion Systems Ltd., Oxford, UK)	RMSD CMC	--	--
Stoeve et al. [65]	Triple-axis linear acceleration and angular velocity	Men; women	Youth; adult	Sub-elite	836	--			✓	IMUs (brand not specified)	200 Hz	A total of 8424 shots and 24,254 passes taken from 10 sessions in a laboratory (controlled exercises) and 10 during training or games (field sessions not following a fixed protocol)	Data compared against labeling made by trained experts using video camera recordings	--	--	F1 score Sensitivity
Wilmes et al. [66]	Three-dimensional acceleration, angular velocity, and magnetic field strength	Men	Adult	Sub-elite	11	--	✓			IMUs (MPU 9150, Invensense, San Jose, CA, USA)	500 Hz	Four short passes (low intensity), four long passes (medium intensity), and four maximum instep kicks (maximal intensity) all interspersed with about 10 s resting	Data compared against and correlated with eight optoelectronic motion cameras (Vicon V5 cameras, Vicon Motion Systems Ltd., Oxford, UK)	ANOVA RMSD CMC	--	--
Yu et al. [67]	Three-dimensional displacements, acceleration, and angular velocity; foot velocity and backwing height	--	--	--	10	--	✓			IMU sensor (ICM-20649, TDK InvenSense MEMS Motion Tracking™ Device)	100 Hz	Instep kicking test	Data compared against “Tracker” using two high-speed cameras positioned in front and side view (brand not specified)	RMSD %Error Bland–Altman plots	--	--

*N*: number of participants; min: minimum; max: maximum; ICC: intraclass correlation coefficient; GLMM: general linear mixed model; % percentage; RMSD: root mean square difference; CMC: coefficient of multiple correlation; -- represents not applicable, information not reported, or unclear.

**Table 3 sensors-24-07912-t003:** Results regarding measurement properties extracted from the studies included in the current review.

Reference	Wearable Device/System	Validity			Reliability			Accuracy	Summary
		Concurrent	Discriminative	Crossed	Between-Session	Within-Session	Intra-Unit	Event Detection	
Bastiaansen et al. [56]	MPU-9150, InvenSense		Knee extension velocity and load *p* = 0.02 ES = 0.94–0.95 Hip load *p* = 0.97 ES = 0.02						Authors claim the discriminative validity of the wearable for the knee load but not for the hip load
Blair et al. [57]	MVN Link, Xsens Technologies	All metrics collected *p* = -- ES = trivial							Authors claim the overall concurrent validity of the wearable
Burland et al. [58]	IMeasureU Blue Trident dual-g, VICON				Cumulative impact load and step count ICC = 0.58–0.87 *p* ≤ 0.056 Cumulative bone stimulus ICC = 0.95–0.96 *p* < 0.001				Authors indicate lower reliability of the wearable to measure ball kicking as compared to other soccer actions
Cuperman et al. [59]	Ivensense MPU-9150							CNN + bLSTM model F1 score = 96.67% Accuracy = 98.3% QDA, kNN, and decision tree models Accuracy = 40–90%	Authors claim better accuracy of the wearable using deep learning processing models as compared to traditional machine learning
de Vries et al. [60]	GT1M/GT3X, ActiGraph							One-axis ANN models Correctly classified activity = 71.4–77.9% Three-axis ANN models Correctly classified activity = 82.1–82.4%	Authors claim better accuracy of the wearable using classification models considering triaxial data as compared to uniaxial
Duncan et al. [61]	GENEActiv, Activinsights			Acc count rho = 0.67–0.75 *p* = 0.0001 AUC = 0.62–0.72 Sensitivity = 59.1–75.3 Specificity = 66.3–68.1					Authors indicate low cross-validity of the wearable output
Lewis et al. [62]	PlayerMaker™/MPU-9150, InvenSense	Ball release velocity r^2^ = 0.96 *p* = --				Ball release velocity CV = 3.93–14.35%			Authors claim the concurrent validity and reliability of the wearable
Marris et al. [63]	PlayerMaker™/MPU-9150, InvenSense	Ball touches and releases PA = 95.1–97.6% SE = 0.0–0.1%					Ball touches and releases CV = 1.8–2.3% SE = 0.0–0.2%		Authors claim the concurrent validity and reliability of the wearable
Steijlen et al. [64]	ICM-20649, InvenSense	Hip and knee angle RMSD = 4–18° CMC = 0.63–0.99 Angular velocity RMSD = 61–103°/s CMC = 0.93–0.99 *p* = --							Authors claim the concurrent validity of the wearable
Stoeve et al. [65]	brand not specified							CNN model F1 score = 0.887–0.928 Sensitivity = 45–84% SVM model F1 score = 0.648–0.815 Sensitivity = 21–39% LSTM and convLSTM models F1 score = 0.777–0.910 Sensitivity = --	Authors claim the accuracy of the wearable using the CNN processing model but not the SVM model; deep learning models outperformed machine learning
Wilmes et al. [66]	MPU 9150, Invensense	Hip and knee angle RMSD = 5–8° CMC = 0.96–0.97 Angular velocity RMSD = 78–177°/s CMC = 0.81–0.89 *p* = --							Authors claim the concurrent validity of the wearable
Yu et al. [67]	ICM-20649, TDK InvenSense	Reconstructed position RMSD = 0.07 m Foot velocity RMSD = 7.47 m/s Error = 4% Backswing height RMSD = 0.74 m Error = 3%							Authors claim the concurrent validity of the wearable

*p* = *p*-value; ES = effect size; ICC: intraclass correlation coefficient; RMSD: root mean square difference; CMC: coefficient of multiple correlation; -- represents information not reported or unclear.

**Table 4 sensors-24-07912-t004:** Ratings of methodological quality for the studies that addressed validity aspects.

Reference	Item 1	Item 2	Item 3	Item 4	Item 5	Item 6	Item 7	Rating
Bastiaansen et al. [56]	Good	Fair	Poor	Fair	Excellent	Excellent	NA	Moderate
Blair et al. [57]	Good	Fair	Poor	Excellent	Excellent	Poor	NA	Moderate
Duncan et al. [61]	Good	Fair	Poor	Fair	Excellent	Excellent	Excellent	Moderate
Lewis et al. [62]	Excellent	Fair	Poor	Excellent	Excellent	Excellent	NA	Moderate
Marris et al. [63]	Good	Fair	Poor	Excellent	Excellent	Poor	NA	Moderate
Steijlen et al. [64]	Good	Fair	Poor	Excellent	Excellent	Excellent	NA	Moderate
Wilmes et al. [66]	Good	Fair	Poor	Excellent	Excellent	Excellent	NA	Moderate
Yu et al. [67]	Good	Fair	Poor	Good	Excellent	Poor	NA	Moderate

Items from the COSMIN checklist (form H); NA: not applicable.

**Table 5 sensors-24-07912-t005:** Ratings of methodological quality for the studies that addressed reliability aspects.

Reference	Item 1	Item 2	Item 3	Item 4	Item 5	Item 6	Item 7	Item 8	Item 9	Item 10	Item 11	Rating
Burland et al. [58]	Good	Fair	Poor	Excellent	Excellent	Excellent	Good	Excellent	Good	Fair	Excellent	Moderate
Lewis et al. [62]	Excellent	Fair	Poor	Excellent	Good	Fair	Good	Fair	Good	Fair	Poor	Moderate
Marris et al. [63]	Good	Fair	Poor	Excellent	Good	Fair	Good	Fair	Good	Fair	Poor	Moderate

Items from the COSMIN checklist (form B). Items 12–14 did not apply to studies included in the present review.

**Table 6 sensors-24-07912-t006:** Ratings of methodological quality for the studies that addressed accuracy aspects.

Reference	Domain 1		Domain 2	Domain 3		Domain 4		Rating
	Item A	Item B	Item A	Item B	Item A	Item B	Item A	
Cuperman et al. [59]	High	Unclear	Low	Unclear	Unclear	Unclear	Unclear	Low
de Vries et al. [60]	High	Unclear	Low	Low	Unclear	Low	Unclear	Moderate
Stoeve et al. [65]	High	Low	Low	Low	Unclear	Low	Unclear	Moderate

Items from the QUADAS-2 tool.

## Data Availability

Data supporting the findings of the current review are primarily derived from previously published studies, which have been cited when pertinent. Additional datasets also containing information that supports our findings are available without undue reservation in with The Open Science Framework (OSF) OSF (https://doi.org/10.17605/OSF.IO/28BDV). The percentage in the similarity report of the manuscript submitted and the latest versions were also made available (respectively 10%—https://doi.org/10.5281/zenodo.14053101 and 12%—https://doi.org/10.5281/zenodo.14286234). This article is a revised and expanded version of a conference paper entitled “Work in progress: reviewing measurement properties of wearables in computing ball kicking features”, which was presented as a poster at the 22nd LACCEI International Multi-Conference for Engineering, Education and Technology, Hybrid, 17–19 July 2024.

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
