# Peer review of "Measurement Properties of Wearable Kinematic-Based Data Collection Systems to Evaluate Ball Kicking in Soccer: A Systematic Review with Evidence Gap Map†"

_sensors, 2024, doi:10.3390/s24247912_

Round 1

Reviewer 1 Report

Comments and Suggestions for Authors

Thank you for the opportunity to review the article, "Measurement properties of wearable kinematic-based data collection systems to evaluate ball kicking in soccer: a systematic review with evidence gap map." In my opinion, from a methodological point of view, this manuscript meets the standards for the preparation of systematic reviews. It also provides a comprehensive and well-organized exploration of the evidence on wearable technologies for the analysis of ball kicking in soccer, a topic of current interest in new technologies and with a lack of research.

In my view, this paper has a clear structure and presentation for several reasons:

- The introduction effectively describes the purpose, methodology, and significance of the study, providing a solid foundation for the systematic review.

- The methodology is rigorous, including the use of tools such as COSMIN and QUADAS-2 for the assessment of methodological quality.

- Data presentation is sufficiently clear and complete, with the abstract including key findings, main characteristics of the included studies, and evidence gaps, which improves the transparency and readability of the manuscript.

- Limitations are identified, highlighting demographic and methodological gaps in the findings presented, which provides a balanced narrative of the findings.

Below are some questions and/or suggestions for improving the manuscript:

1. Line 130: "Inclusion criteria. Studies were included (...) (iv) with full-text available – in the English language for download".

Does this mean that only open access articles have been included? If so, important works from very prestigious journals could be excluded. Please clarify this issue and point out, if necessary, this limitation.

2. In the inclusion criteria (lines 127-137) it states "(vi) participants as soccer players and/or human data collected in a football related environment". Does this mean that studies in similar sports such as 7-a-side football, indoor football or beach football have not been included? As before, please clarify this issue and point out, if necessary, this limitation.

3. Although your work focuses on the evaluation of ball kicking in soccer, the discussion could be enriched by studies that apply wearables in a very similar way, to analyze upper body throws or kicks, in team sports such as handball, basketball or volleyball. Advantages and limitations observed in other studies (including other systematic reviews) with wearables could certainly suggest advances for the evaluation of ball kicking in soccer.

4. Overal, expand the discussion to include practical recommendations for the use of wearable devices in training and competition. Discuss how coaches and practitioners could implement the findings, considering the limitations of the devices.

Reviewer 2 Report

Comments and Suggestions for Authors

This article systematically reviews the measurement properties (validity, reliability, and accuracy) of wearable devices in assessing soccer kicking actions. It integrates evidence from seven databases and follows the PRISMA 2020 guidelines for analysis. The study highlights that certain wearable devices, such as MPU-9150 and PlayerMaker™, demonstrate moderate validity in controlled environments, showcasing their potential for analyzing kinematic variables in soccer. However, the evidence for reliability and accuracy, especially in real-game scenarios, remains limited. While the research is rigorous and well-structured, there are opportunities for improvement to enhance both its academic and practical relevance.

1.     Although the article focuses on validity, it provides limited insights into reliability and accuracy, which are crucial for practical applications. Expanding on the consistency of devices across time periods and experimental settings could address this gap. Additionally, discussing potential biases arising from experimental conditions would strengthen the study’s implications.

2.     The reviewed studies primarily involve adult male soccer players, limiting the generalizability of the findings. Including perspectives on underrepresented groups, such as female, youth, and amateur players, would enhance the breadth and applicability of the conclusions. It is recommended to highlight this limitation explicitly and advocate for broader sample inclusion in future studies.

3.     The figures, while informative, lack sufficient explanations in some cases. For example, Figures 6 and 7, which present interactions between measurement properties and wearable devices, could benefit from additional textual descriptions to help readers understand the findings more intuitively.
